# Efficacy of Needle-Less Intradermal Vaccination against Porcine Epidemic Diarrhea Virus

**DOI:** 10.3390/pathogens10091115

**Published:** 2021-08-31

**Authors:** SeEun Choe, Gyu-Nam Park, Sok Song, Jihye Shin, Van Phan Le, Van Giap Nguyen, Ki-Sun Kim, Hye Kwon Kim, Bang-Hun Hyun, Dong-Jun An

**Affiliations:** 1Virus Disease Division, Animal and Plant Quarantine Agency, Gimchen, Gyeongbuk-do 39660, Korea; ivvi59@korea.kr (S.C.); changep0418@gmail.com (G.-N.P.); ssoboro9@gmail.com (S.S.); shinjibong227@gmail.com (J.S.); kisunkim@korea.kr (K.-S.K.); hyunbh@korea.kr (B.-H.H.); 2Department of Veterinary Microbiology and Infectious Diseases, Faculty of Veterinary Medicine, Vietnam National University of Agriculture, Hanoi 100000, Vietnam; letraphan@gmail.com (V.P.L.); nvgiap@vnua.edu.vn (V.G.N.); 3Department of Microbiology, College of Natural Sciences, Chungbuk National University, Cheongju 28644, Korea; khk1329@chungbuk.ac.kr

**Keywords:** porcine epidemic diarrhea virus (PEDV), needle-less, intradermal, vaccine, piglet, colostrum

## Abstract

To prevent diarrhea in suckling piglets infected by porcine epidemic diarrhea virus (PEDV), porcine epidemic diarrhea (PED) vaccines are administered mainly through intramuscular (IM) or oral routes. We found that growing pigs vaccinated with an inactivated PEDV vaccine via the intradermal (ID) route had higher neutralizing antibody titers and cytokine (IFN-γ, IL-4, and IL-10) levels than non-vaccinated pigs. In addition, suckling piglets acquired lactogenic immunity from pregnant sows inoculated with an ID PED vaccine. We evaluated the efficacy of vaccination via this route, along with subsequent protection against virulent PEDV. At six days post-challenge, the survival rate of suckling piglets exposed to virulent PEDV was 70% for the ID group and 0% for the mock group (no vaccine). At necropsy, villi length in the duodenum and ileum of piglets with lactogenic immunity provided by ID-vaccinated sows proved to be significant (*p* < 0.05) when compared with those in piglets from mock group sows. Thus, vaccination using an inactivated PED vaccine via the ID route provides partial protection against infection by virulent PEDV.

## 1. Introduction

Porcine epidemic diarrhea (PED) is a highly infectious and contagious enteric disease of swine [1]. The disease is characterized by severe diarrhea, vomiting, dehydration, and death; the mortality rate in suckling piglets is >90% [2]. Porcine epidemic diarrhea virus (PEDV), the causative agent of PED, is an RNA virus belonging to the alpha genus of the coronavirus family [3]. 

Levels of PEDV-specific sIgA antibody in colostrum and milk are key to determining the extent of passive immunity against virulent PEDV infection afforded to piglets [1]. Recently, pregnant sows on pig farms in Korea have received a live attenuated oral PED vaccine to increase production of IgA and IgG antibodies; the initial vaccination is followed by two additional doses of the inactivated PED vaccine. This strategy is aimed at providing piglets with high levels of sIgA and IgG via colostrum and milk. 

Currently, commercial inactivated and live attenuated PED vaccines are used to induce PED antibodies in pregnant sows in Korea. The main vaccination route is intramuscular (IM); the secondary route is oral (for the live attenuated vaccine). Due to needle-free administration, an intradermal (ID) vaccine is more animal-friendly and prevents accidental transmission of pathogens through reuse of needles; it also reduces the risk of broken needles being left in the muscle [4]. Compared with IM injection, vaccination into the dermis has the advantage of accessibility to dendritic cells at the site of administration, along with close proximity to skin-draining lymph nodes; this results in a more rapid and direct response to the antigen in the vaccine. ID vaccination of animals has been used to protect pigs against Aujeszky’s disease [5,6,7,8], porcine reproductive respiratory syndrome virus (PRRSV) [9], and foot-and-mouth disease virus (FMDV) [10]. In humans, ID vaccination protects against influenza [11,12,13], rabies [14,15,16], and hepatitis B [17,18]. 

Here, we examined the possibility of using ID vaccination as a novel route of administration for an inactivated PED vaccine. Immune responses and cytokine production by pigs inoculated via the ID route were analyzed. In addition, pregnant sows were inoculated with the PED vaccine via the ID route, after which piglets were inoculated with a virulent PEDV to examine the level of protection afforded by the vaccine. 

## 2. Results

### 2.1. Selection of the Adjuvant

Pigs (50 days old) were vaccinated against PED by ID route; the adjuvants used were Montanide^TM^ IMS 1313 VG NPR (SEPPIC, Paris, France), Montantide^TM^ GEL 01 PR (SEPPIC, Paris, France), or Carbopol 940 polymer (NF) (Lubrizol, Wickliffe, OH, USA). ID injections were performed using a Pulse 250 apparatus (Pulse NeedleFree Systems, Inc., Lenexa, KS, USA) (Figure 1). 

Edema was noted at the site of injection after PED vaccination using Carbopol 940, which disappeared after one day. Edema caused by injection of Montanide^TM^ IMS 1313 and GEL 01 PR disappeared after two to three days (Figure 1C). Pigs vaccinated ID with the IMS 1313 adjuvant showed antibody titers of 5.67 log_2_ at 28 days post-vaccination (dpv) and 4.33 log_2_ at 56 dpv (Figure 2A). Pigs inoculated ID with GEL 01 PR had titers of 5.33 log_2_ at 28 dpv and 3.33 log_2_ at 56 dpv (Figure 2A). Pigs inoculated ID with Carbopol 940 maintained titers of 5.67 log_2_ at 28 dpv and 3.67 log_2_ at 56 dpv (Figure 2A). 

After ID inoculation with the inactivated PED vaccine (plus the Carbopol 940 adjuvant), relative levels of interferon γ (IFN-γ) were significantly higher (0.63 at 1 dpv, 0.87 at 8 dpv, 1.50 at 15 dpv, 1.57 at 22 dpv, 0.97 at 36 dpv, and 0.57 at 50 dpv) than those in the mock group (no vaccine) (0.15 at 1 dpv, 0.14 at 8 dpv, 0.19 at 15 dpv, 0.15 at 22 dpv, 0.12 at 36 dpv, and 0.17 at 50 dpv) (*p* < 0.01) (Figure 2B). Relative interleukin 4 (IL-4) levels were significantly higher in pigs vaccinated ID (1.07 at 22 dpv and 0.83 at 36 dpv) than in the mock group (0.3 at 22 dpv and 0.31 at 36 dpv) (*p* < 0.01) (Figure 2C). Finally, relative interleukin 10 (IL-10) levels were significantly higher in pigs vaccinated ID (0.77 at 15 dpv, 1.60 at 22 dpv, and 0.8 at 36 dpv) than in the mock group (0.3 at 15 dpv, 0.32 at 22 dpv, and 0.27 at 36 dpv) (*p* < 0.05 and *p* < 0.01) (Figure 2D). 

### 2.2. Antibody Titers and Safety of Pregnant Sows after ID Inoculation 

After two ID injections of inactivated PED vaccine (plus the Carbopol 940 adjuvant) into four pregnant sows, the number of live births averaged 11.7. The number of live births for the three pregnant sows in the mock group (no vaccine) averaged 11.3. There were no abortions and no clinical signs such as diarrhea (Table 1) in any of the pregnant sows. After two vaccinations, the average neutralizing antibody (NA) titer at the time of delivery was 5.75 log_2,_ and the NA titer in colostrum was 5.25 log_2_ (Table 1). Titers in the mock group were 2.3 log_2_ at delivery, whereas those in colostrum were 1.6 log_2_ (Table 1). The IgA titer in colostrum was higher in the vaccination group (O.D. = 1.25) than in the mock group (O.D. = 0.48) (Table 1).

### 2.3. Diarrhea Score and Weight Changes in Suckling Piglet Challenged with Virulent PEDV

After challenging suckling piglets (five days old) with virulent PEDV, mock group piglets showed evidence of diarrhea from 20 h post-challenge, followed by severe diarrhea from two days post-challenge (dpc). By contrast, suckling piglets that ingested colostrum showed only mild diarrhea. Sucking piglets (five days old) of vaccinated sows showed low diarrhea scores; the averages were 1.0 (for ID1), 0.33 (for ID2), 1.33 (for ID3), and 0.67 (for ID4). However, mock suckling piglets showed higher diarrhea scores (average = 3 at 3 dpc) (Figure 3A). The body weight of suckling piglets that ingested colostrum from vaccinated sows decreased until about 5 dpc, but recovered gradually from 6 dpc (Figure 3B). 

### 2.4. Survival and Viral RNA Copy Number in Suckling Piglets

After challenging with virulent PEDV, the survival rate of sucking pigs born to vaccinated sows was 100% at 3 dpc, 91.6% at 4 dpc, 83.3% at 5 dpc, and 66.6% at 6 dpc (Figure 4A). However, the survival of suckling piglets borne to mock group sows was 88.8% at 2 dpc, 77.7% at 3 dpc, 33.3% at 4 dpc, 11.1% at 5 dpc, and 0% at 6 dpc (Figure 4A). The average serum titer of NA in one-day-old piglets was higher (5.41 log_2_) in those whose mothers received ID vaccination (ID sows nos. 1–4) than in those whose mothers were in the mock group (1.44 log_2_) (MO sows nos. 1–3) (Figure 4B). The number of PEDV RNA copies in feces at 2 dpc was much higher in the piglets of mock group sows (6.33 log_10_ in piglets from sow no. MO1, 5.43 log_10_ in piglets from sow no. MO2, and 6.29 log_10_ in piglets from sow no. MO3) than in the piglets from vaccinated sows (3.78 log_10_ for piglets from sow no. ID1, 0.00 log_10_ for piglets from sow no. ID2, 1.03 log_10_ for piglets from sow no. ID3, and 1.43 log_10_ for piglets from sow no. ID4) (Figure 4C). 

### 2.5. Morphological Changes in the Small Intestine of Suckling Piglets

Necropsy of piglets challenged with PEDV showed significant vaccination-dependent differences in villus height (VH) in the duodenum and ileum (*p* < 0.05) (Figure 5A). 

However, there was no difference in crypt depth (CD) in the duodenum, jejunum, and ileum between piglets of mock group sows and those of inoculated sows (Figure 5A). There was also no significant difference in the VH/CD ratio in the duodenum, jejunum, and ileum (Figure 5B). However, the small intestine epithelial cells in the duodenum and ileum of mock group piglets (Figure 6A,C) were more damaged than those of piglets (Figure 6D,F) from vaccinated sows.

## 3. Discussion

Intradermal vaccination has the advantage of targeting antigen-presenting cells (APCs) in the epidermis, which reside in close proximity to skin-draining lymph nodes. This is important because the skin harbors many more specialized APCs, which are essential for effective induction of cell-mediated immunity [19,20,21], than muscle tissue. This means that the amount of adjuvant and/or antigen in an intradermal vaccine can be lower, yet still achieve comparable or even greater efficacy [22]. ID is proposed as the route of administration for vaccines against Aujeszky’s disease because they induce a stronger immune response and greater protection against challenge than vaccines delivered via the IM route [23,24,25]. An attenuated PRRS vaccine administered via the ID route can efficiently induce a protective immune response in pigs subsequently exposed to a genetically diverse PRRSV isolate [9]. The safety and efficacy of a new intradermal one dose vaccine containing the PCV2 antigen (Porcilis^®^ PCV ID) was evaluated and showed that the vaccine was safe and provided protection until at least 23 weeks post-vaccination, which is a typical slaughter age [4].

Here, we found that the Carbopol 940 polymer (NF) was the best adjuvant for the inactivated PED vaccine because it generated a sufficient and persistent NA titer, with fewer side effects. The anti-PED NA titer and persistence induced by the IMS 1313 adjuvant was slightly better than that in pigs injected with the Carbopol 940 polymer adjuvant, but there was prolonged edema at the injection site (lasting for two to three days). Pigs inoculated with the PED vaccine plus the Carbopol 940 adjuvant via the ID route had significantly higher IFN-γ, IL-4, and IL-10 levels than mock pigs (no vaccine). There was a significant difference between the levels of the three cytokines in ID PED vaccine pigs and mock pigs at 1–50 dpv (for IFN-γ), 22–36 dpv (for IL-4), and 15–36 dpv (for IL-10). Many studies report that ID vaccination results in stronger cell-mediated immune responses against hepatitis B virus [26], influenza virus [27], and human immunodeficiency virus (HIV) [28] than IM vaccination. Anti-influenza vaccine via the ID route in humans induced both effector CD4 and CD8 T cell responses, whereas IM injection induced strong effector CD4 in the absence of CD8 T cells [27]. The safety and cellular immunogenicity of HIV-lipopeptide candidate vaccine (LIPO-4) via the ID route was well tolerated, required one-fifth of the intramuscular dose, and induced similar HIV-specific T cells (CD4+ or CD8+) responses [28]. IFN-γ-secreting cells reached maximum levels and upregulation of IFN-γ gene expression in PBMC was also detected in ID group pigs after administration of the booster dose by Aujeszky’s vaccine [29]. Another study that compared IM foot-and-mouth disease (FMD) vaccination with ID FMD vaccination suggested that the latter protects animals when their serum neutralization antibody titers are low [30]. Data suggests that this effect is partly associated with humoral immunity because ID vaccination is a major contributor to cell-mediated immune responses that induce mobilization of inflammatory dendritic cells [31,32]. 

The PEDV RNA copy number in the feces and the diarrhea score of piglets born to vaccinated sows in this study were lower than those in piglets born to mock group sows. The weight of piglets receiving lactogenic immunity fell after challenge with virulent PEDV, but recovered from 6 dpc. There were significant differences in VH in the duodenum and ileum between suckling piglets with lactogenic immunity induced by ID vaccination and those without (*p* < 0.05). In both the onset of immunity and duration of immunity challenge studies with PCV2 or *M. hyopneumoniae* vaccine via the ID route, significant reductions of the PCV2 load in lymphoid tissue, lungs, serum and fecal swabs and *M. hyopneumoniae*-induced lung lesions were observed [4]. In addition, a significant positive effect on average daily weight gain (between 44 and 59 g/day) in the finishing phase was observed from pigs administered with the PCV2 vaccine via the ID route [4]. The survival rate after challenge of newborn suckling piglets delivered by sows vaccinated with ID PED vaccine in this study was similar to that of piglets delivered by sows vaccinated with PED aP2 subunit vaccine (loaded hydroxypropyl methylcellulose phthalate microspheres plus RANKL-secreting *L. lactis*) [33]. 

In conclusion, ID vaccination of sows with an inactivated PED vaccine provided suckling piglets with partial protection against challenge with virulent PEDV. In the future, a commercial inactivated PED vaccine will be compared with respect to its ability to protect suckling piglets via the IM and ID routes. In addition, it is necessary to conduct a comparative experiment that involves oral administration with a commercial live attenuated PED vaccine.

## 4. Materials and Methods

### 4.1. Adjuvant Selection and Measurement of Cytokine Levels in Growing Pigs 

The PEDV vaccine candidate strain (SGP-M1) was derived from piglets with watery diarrhea in the Gimpo region of South Korea. The virus was isolated by passaging 50 times in a Vero cell line in the presence of porcine trypsin (5 μg/mL). The SGP-M1 strain (10^7.0^ TCID_50_/mL) was exposed to 5% binary ethylenimine (BEI) at room temperature for 48 h to inactivate the virus, followed by dialysis to neutralize the BEI. Since the commercial inactivated PED vaccine in Korea has a minimum titer of 10^7.0^ TCID_50_/mL, the ID inoculation dose used in this experiment was set at a similar value. For the optimal efficacy of the ID PED vaccine, the suitable adjuvant selected through a growing pig examination for three adjuvants (Montanide^TM^ IMS 1313 VGN, Montanide^TM^ GEL 01 PR, or Carbopol 940 polymer), and the selected adjuvant was used for an ID PED vaccine for the pregnancy sow examination. 

All pigs used in this study obtained from a farm were confirmed as free from respiratory viruses (swine influenza virus (SIV), PRRSV, porcine circovirus 2 (PCV2)) and digestive viruses (PEDV, transmissible gastroenteritis virus (TGEV), porcine rotavirus (PRoV)). In addition, pigs were tested negative for the viruses listed above. A total of 11 healthy pigs (50 days old) were used for adjuvant selection and assessment of immunogenicity. For adjuvant selection and assessment of immunogenicity and safety, nine pigs (50 days old) were inoculated via the ID route with different adjuvants (Montanide^TM^ IMS 1313 VGN, Montanide^TM^ GEL 01 PR, or Carbopol 940 polymer). For each of the three adjuvants, three groups of pigs (n = 3/group) were vaccinated twice via the ID route (0.5 mL/pig; 10^7.0^ TCID_50_/mL) at intervals of 2 weeks. Two pigs (50 days old) were assigned to the mock group (no vaccine) (Figure 2A). ID injection was performed using the Pulse 250 apparatus (Pulse NeedleFree Systems, Inc., KS, USA) (Figure 1A) and observed the injection site at 1 h (Figure 1B) and 3 days (Figure 1C) post-intradermal inoculation into the neck muscle. Blood samples were collected at days 0, 7, 14, 21, 28, 35, 42, 49, and 56 dpv (i.e., 1 week intervals) to examine seroconversion (Figure 2A). 

To measure IFN-γ, IL-4, and IL-10, eight healthy pigs (65 days old) without PED antibodies were used. Four pigs (65 days old) were inoculated twice via the ID route (0.5 mL/pig; 10^7.0^ TCID_50_/mL) with Carbopol 940 adjuvant at intervals of 2 weeks, and four pigs (65 days old) were not vaccinated (mock group) (Figure 2B–D). Blood samples were collected at 0, 1, 8, 15, 22, 36 and 50 dpv (Figure 2B–D).

#### 4.1.1. NA Test

PEDV NA titers in pig serum were measured as described previously [34]. Briefly, Vero cells were maintained in a-MEM (Gibco, New York, NY, USA) supplemented with 5% fetal bovine serum and 2% antibiotic-antimycotic agent (AA) mixture (Gibco, New York, NY, USA). Serum was inactivated at 56 °C for 30 min and stored at 20 °C. Serum was diluted 2-fold (starting from the first well of a 96-well plate). Next, virulent PEDV (SGP-M1 strain; 2 × 10^2^ TCID_50_/0.1 mL) was reacted with an equal volume (50 μL) of pig serum in the 96-well plate for 1 h at 37 °C. The mixture (virus and serum) was then inoculated into a 96-well plate containing Vero cells and incubated at 37 °C for 2 h. The plate was washed three times with phosphate-buffered saline (PBS, pH 7.2), and 100 μL of infection medium (a-MEM, 2% AA, 0.3% tryptose phosphate broth (Sigma-Aldrich, St. Louis, MO, USA), 0.02% yeast extract (Gibco, New York, NY, USA), and 2.5% Trypsin (2 μg/mL) (Gibco, New York, NY, USA)) was added to each well. The plates were placed in a 5% CO_2_ incubator at 37 °C and observed 3 days later for cytopathogenic effects (CPE). PED NA titers were expressed as the reciprocal of the highest serum dilution that inhibited CPE.

#### 4.1.2. Isolation of Porcine Peripheral Blood Mononuclear Cells and cDNA Synthesis

Density gradient centrifugation on Lymphoprep^TM^ solution (STEMCELL Technologies, Canada) was used to isolate peripheral blood mononuclear cells (PBMCs) from 15 mL of porcine blood collected in ethylene-diamine-tetraacetic acid as an anticoagulant [35]. Briefly, anticoagulated blood was diluted in PBS and layered gently over an equal volume of Lymphoprep^TM^ solution (1:1, *v*/*v*) in a conical tube, which was then centrifuged for 20 min at 500× *g*. The mononuclear cell layer (leucocyte phase) was collected gently and washed with PBS. The viability of the purified PBMCs was assessed by Trypan blue staining (Sigma-Aldrich, St. Louis, MO, USA). Total RNA was extracted from each PBMC sample using an RNeasy mini kit (QIAGEN Inc., Germantown, MD, USA) and eluted with 50 μL of elution buffer. The purity and concentration of each RNA sample was assessed using a BioDrop spectrometer (Biochrom Ltd., Cambourne, UK). Total RNA (200 ng) was reverse-transcribed using a HelixCript^TM^ easy cDNA synthesis kit (NanoHelix Co., Ltd., Daejeon, Korea). 

#### 4.1.3. Quantification of Cytokine Gene Expression by PBMCs

To measure expression of mRNA encoding IFN-γ, IL-4, and IL-10, quantitative real-time PCR (qRT-PCR) on a C1000 thermocycler (Bio-Rad, Hercules, CA, USA) was performed using the SsoAdvanced Universal Probes Supermix (Bio-Rad, Hercules, CA, USA) and gene-specific primers (Cosmo Genetech Co., Seoul, Korea) [36,37] (Table 2). The housekeeping gene, glyceraldehyde-3-phosphate dehydrogenase (GAPDH), was selected as an endogenous control. The cDNA (4 μL/20 μL) was used as a template for qRT-PCR. The thermal profile for the amplification was as follows: 95 °C for 30 s, followed by 40 cycles of 95 °C for 10 s and 60 °C for 30 s. Negative control and no-template control (NTC) duplicates were included in each experiment. Data analysis was performed by Bio-Rad CFX Manager software (version 3.1). NTC controls were determined as negative and reliable if the quantification cycle (Cq) was ≥35. Data were analyzed using the 2^−∆∆Ct^ method [38], in which the levels of each cytokine, normalized to that of GAPDH cDNA, were expressed as relative quantities compared with those in control pigs.

### 4.2. Vaccination by ID Route to Pregnant Sows 

Four of seven healthy pregnant sows obtained from a farm free from specific diseases (SIV, PRRSV, PCV2, PEDV, TGEV, PRoV) were inoculated with the inactivated PED vaccine (plus Carbopol 940 polymer adjuvant) by ID injection. The remaining three pregnant sows were assigned to the mock group (no vaccine). Pigs received two doses of inactivated PED vaccine via the ID route (0.5 mL/pig; 10^7.0^ TCID_50_/mL): one at 5 weeks and one at 2 weeks before delivery. In addition, blood (for measurement of NA titers) was collected before vaccination and again immediately after delivery. Colostrum was also collected immediately after delivery. The number of live, dead, and aborted piglets was counted during the observation period.

### 4.3. Challenge of Piglets with Virulent PEDV after Ingestion of Maternally-Derived Antibodies 

Suckling piglets were housed with their sows. Healthy 5-day-old piglets were selected at random from each farrowing sow (three piglets per sow) in the vaccinated and mock groups; these piglets were then inoculated with virulent PEDV. A total of 21 sucking piglets (12 piglets in the vaccinated group and 9 in the mock group) were transported from the farm to an environmentally controlled facility, where they received artificial milk four times daily. The suckling piglets (aged 5 days) were challenged orally with 10 mL (10^3.0^ TCID_50_/mL) per piglet of virulent PEDV G2b type. Clinical signs of diarrhea, morbidity, and mortality were monitored three times daily for up to 8 dpc. Rectal swabs were collected and body weight was checked daily (from day 0 (pre-challenge) to 8 dpc). Diarrhea was scored daily during the observation period (8 days). Fecal consistency was scored as follows: 0, solid; 1, pasty; 2, semi-liquid; and 3, completely liquid. If piglets died after virulent PEDV challenge, autopsies were performed immediately; the remaining surviving piglets were euthanized at 8 dpc.

#### 4.3.1. QRT-PCR of Fecal Samples from Piglets

QRT-PCR was performed to detect the PED antigen copy number in fecal swab samples to assess PEDV shedding post-challenge. The AnyQ PEDV qRT-PCR (MEDIAN Diagnostic Inc., Cat No. PS105, Chuncheon, Korea), which uses TaqMan probes to detect the PEDV spike region, was used in accordance with the manufacturer’s instructions. The qRT-PCR program comprised the following steps: cDNA synthesis (50 °C, 30 min) and initial inactivation (95 °C, 15 min), followed by two-step PCR comprising 42 cycles of denaturation (95 °C, 10 s) and annealing/extension (60 °C, 60 s). 

#### 4.3.2. Detection of PED IgA Antibodies and Histopathological Examination 

IgA levels in the colostrum were tested using the PED IgA ELSIA kit (BioNote Inc., Cat No. EB4410PO, Hwaseong, Korea). The mean OD_450_ of the standard negative (NCx) and standard positive (PCx) controls was ≤0.200 and ≥0.500, respectively. Interpretation of the results is based on a cut-off value, which was calculated as follows: cut-off value = [0.35 + NCx]. The HV/CD ratio in the jejunum and ileum was measured (six villi per intestinal section). Moribund piglets that showed signs of anorexia and dehydration over a period of 24 h were euthanized on the instruction of institutional veterinarians.

### 4.4. Statistical Analysis 

All statistical analyses were performed using GraphPad Prism software, version 6.0, for Windows. Data were analyzed using one-way analysis of variance followed by Tukey’s multiple-comparison test, and by two-way analysis of variance followed by Bonferroni post-tests. Data are expressed as the mean ± standard error of the mean (SEM), and significant differences (*p* < 0.05 and *p* < 0.01) are indicated by asterisks.

## 5. Conclusions

Administration of an inactivated PED vaccine plus Carbopol 940 polymer adjuvant via the ID route generated higher NA titers and significantly higher levels of cytokine (IL-4, IL-10, and IFN-γ) gene expression than no vaccination (mock). In addition, piglets ingesting colostrum and milk from sows receiving an ID inoculation were provided with partial protection against challenge with virulent PEDV.

## Figures and Tables

**Figure 1 pathogens-10-01115-f001:**
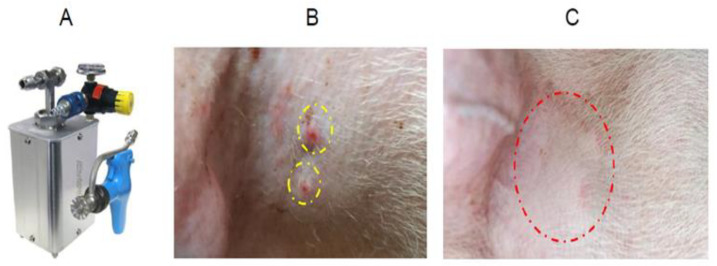
Intradermal inoculation into the neck. The Pulse 250 (needle-free system) intradermal machine (**A**). Photograph of the injection site at 1 h (**B**) and 3 days (**C**) post-intradermal inoculation into the neck muscle.

**Figure 2 pathogens-10-01115-f002:**
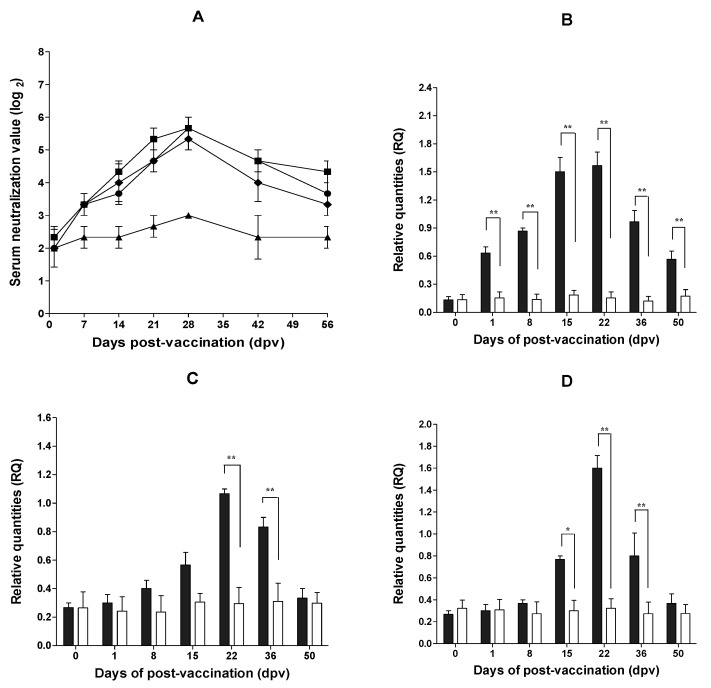
Changes in serum neutralizing antibody titers and cytokine levels after intradermal (ID) inoculation with a PED vaccine. Changes in antibody titers after injection with the three adjuvants (**A**). The Montanide^TM^ IMS 1313 VGN, Montanide^TM^ GEL 01 PR, and Carbopol 940 polymer adjuvants are denoted by squares, rhombuses, and circles, respectively. Mock inoculation is denoted by triangles. Data represent the mean ± SEM. Changes in IFN-γ (**B**), IL-4 (**C**), and IL-10 (**D**) concentrations after inoculation with the inactivated PED vaccine plus the Carbopol 940 polymer adjuvant via the ID route are shown. The intradermal and mock (no vaccine) injections are denoted by gray bars and white bars, respectively. *, *p* < 0.05; **, *p* < 0.01.

**Figure 3 pathogens-10-01115-f003:**
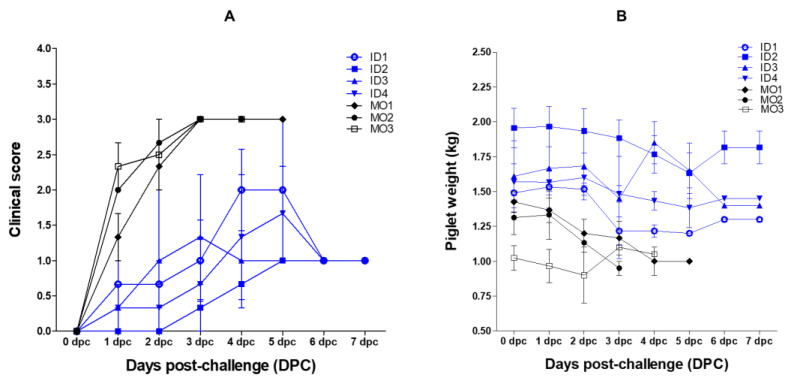
Clinical scores and body weight of piglets inoculated with virulent PEDV. Clinical score for diarrhea (**A**) and changes in body weight (**B**) after challenge with virulent PEDV. Piglets born to sows (nos. ID1, ID2, ID3, and ID4) inoculated intradermally with the PED vaccine plus the Carbopol 940 polymer adjuvant, and piglets born to sows in the mock group (nos. MO1, MO2, and MO3), are denoted by blue and black lines, respectively. Data represent the mean ± SEM.

**Figure 4 pathogens-10-01115-f004:**
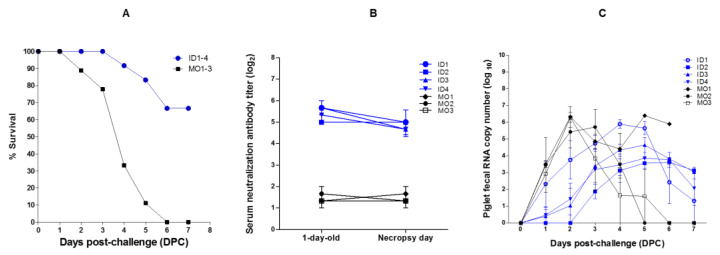
Survival, serum neutralizing antibody titer, and RNA copy number in piglets inoculated with virulent PEDV. Percentage survival (**A**), changes in serum neutralizing antibody titers (**B**), and PEDV RNA copy number in feces of piglets challenged with virulent PEDV (**C**). Piglets born to sows (nos. ID1, ID2, ID3, and ID4) inoculated intradermally with the PED vaccine plus the Carbopol 940 polymer adjuvant, and piglets born to mock group sows (nos. MO1, MO2, and MO3), are denoted by blue and black lines, respectively. Data represent the mean ± SEM.

**Figure 5 pathogens-10-01115-f005:**
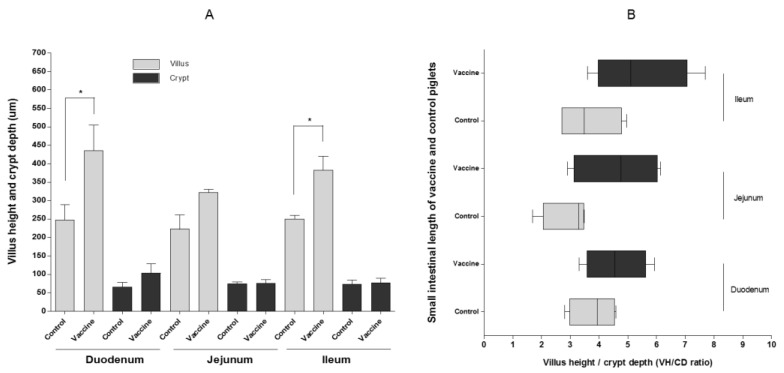
Villus height and length of the small intestine in piglets inoculated with virulent PEDV. Villus and crypt depth in the duodenum, jejunum, and ileum (**A**), and villus height versus crypt depth (VH/CD ratio) (**B**). Data represent the mean ± SEM. * *p* < 0.05.

**Figure 6 pathogens-10-01115-f006:**
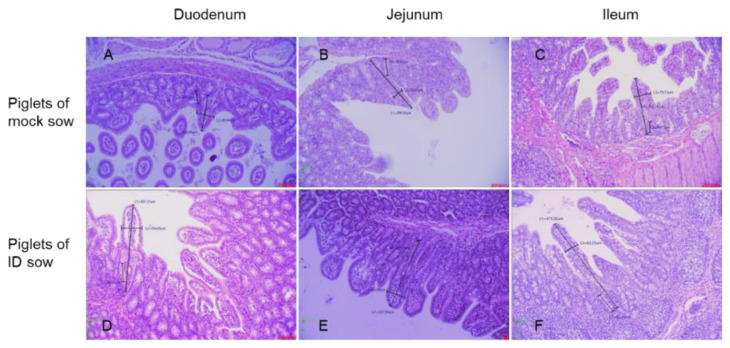
Changes in the morphology of the villus and crypts in the duodenum (**A**), jejunum (**B**), and ileum (**C**) of piglets (from no vaccine sows) inoculated with virulent PEDV. Villus and crypts in the duodenum (**D**), jejunum (**E**), and ileum (**F**) of piglets (from ID PED vaccine sows) inoculated with virulent PEDV.

**Table 1 pathogens-10-01115-t001:** NA titer, IgA ratio, and number of offspring produced by pregnant sows.

Group	Sow	Period of Pregnancy (no. of Piglets)	No. of Live Piglets	No. of Dead Piglets	No. of Abortions	Clinical Signs	NA ^a^ Titer (log_2_)			IgA ELISA (O.D. Value) for Colostrum
							DPV0	Delivery	Colostrum	
Mock	MO1 MO2 MO3	116 (12) 114 (13) 116 (10)	11 13 10	1 0 0	0 0 0	no no no	2 3 4	3 2 2	2 1 2	0.52 0.45 0.49
Inactivated PED (ID)	ID1 ID2 ID3 ID4	114 (13) 116 (10) 114 (11) 115 (14)	13 10 10 14	0 0 1 0	0 0 0 0	no no no no	3 4 2 3	6 6 6 5	5 6 5 5	1.1 1.7 0.8 1.4

^a^ neutralizing antibody. IgA ELISA cut-off value = 0.54.

**Table 2 pathogens-10-01115-t002:** Sequences of the cytokine-specific primers and probes.

Cytokine	Specific Primer Sets (5′→3′)	Probe	Length (bp)	Reference
*IFN-γ*	F-CGATCCTAAAGGACTATTTTAATGCAA R-TTTTGTCACTCTCCTCTTTCCAAT	ACCTCAGATGTACCTAATGGTGGACCTCTT	102	[36]
*IL-4*	F-GTCTGCTTACTGGCATGTACCA R-GCTCCATGCACGAGTTCTTTCT	CCACGGACACAAGTGCGACATCACCTTAC	117	[37]
*IL-10*	F-CGGCGCTGTCATCAATTTCTG R-CCCCTCTCTTGGAGCTTGCTA	AGGCACTCTTCACCTCCTCCACGGC	89	[36]
*GAPDH*	F-ACATGGCCTCCAAGGAGTAAGA R-GATCGAGTTGGGGCTGTGACT	CCACCAACCCCAGCAAGAGCACGC	105	[37]

FAM 5′ was used for IFN-γ, IL-4, IL-10, and GAPDH; BHQ (Black Hole Quencher)-1 was used to target the 3′ end of IFN-γ, IL-4, IL-10, and GAPDH. F: forward primer; R: reverse primer.

## Data Availability

Not applicable.

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
