# Peer review of "Efficacy of Needle-Less Intradermal Vaccination against Porcine Epidemic Diarrhea Virus"

_pathogens, 2021, doi:10.3390/pathogens10091115_

Round 1
Reviewer 1 Report
The manuscript “Efficacy of needle-less intradermal vaccination against porcine epidemic diarrhea virus,” by Choe, et. al., describes two studies in which the authors investigate the consequences of an intradermal vaccination with an inactivated whole virus PEDV vaccine. The first study describes the inoculation of pigs with via IM or ID route and the use of 3 different adjuvants with the PEDV vaccine. The second study evaluates the efficacy of ID vaccination using one adjuvant in sows to provide protection to their suckling piglets following wild-type PEDV challenge.
In general, the authors do convey their message to the reader, but there is a lack of details in the manuscript. The lack of details is to an extent where I cannot determine how they made the decisions they did make.
- It is not clear what “dose” was used for each adjuvant, and why were they picked the dose, or the adjuvants to be used?
- A decision was made to use the Carbopol 940 adjuvant for the sow study, the adjuvant selection was based on the combination of the Carbopol 940 ID vaccination in pigs generating “the highest NA titer and persistence, and had fewer side effects.” Based on figure 2A, as I understand it, the NA titer of the Carbopol 940 ID vaccination was not the highest or had the longest duration of an elevated NA titer. Unless I missed it, there are no numbers provided to support the author’s claim of less edema following ID vaccination with Carbopol 940 compared to the other two adjuvants.
- I may have missed it as well, but there is no statistical support for any differences in the NA titers or cytokine values for any treatment.
- There is a lack of details on how methods were done, e.g., how were the NA titers determined? No details or reference on how to conduct the test was cited. A reader cannot reproduce the studies in this manuscript because of a lack of details. Is there any real difference between titers?
- Just looking at Figures 2 and 3, not much evidence that ID is superior to IM. This may be acceptable since part of the goal is to evaluate ID vaccination in sows.
- For the sow study, there is no IM group. The authors do show that the ID vaccination in sows was helpful, it would have been nice to compare the IM to the ID vaccination in sows.
- In final Discussion paragraph the authors speculate ID vaccination will provide good protection if given after an oral vaccination with attenuated PEDV vaccine. I believe this is too much speculation, and not appropriate.
- There is no description of what a mock vaccination was and what a mock challenge was.
- Overall, the manuscript read well, but a few phrases or words should be revised. For example, in first sentence of the Abstract, I suggest changing sucking to suckling, and epidemical to epidemic.
- The Discussion section should be reviewed for precision. In second paragraph, the sentence beginning with “A previous study showed that………. against virus challenge” I am not sure which previous study is being referred to for which pig disease. The citation I could not find without purchasing the book/journal so I do not know what it states. It could be the cited reference has something to do with the general concept of “ID vaccination triggers other immune mechanisms” which may be true, but this comment is very broad and it may reflect some other example in the citation that has nothing to do with pigs, I do not know. I believe other statements in the Discussion may be a bit broad as well.
Author Response
Reviewer 1
The manuscript “Efficacy of needle-less intradermal vaccination against porcine epidemic diarrhea virus,” by Choe, et. al., describes two studies in which the authors investigate the consequences of an intradermal vaccination with an inactivated whole virus PEDV vaccine. The first study describes the inoculation of pigs with via IM or ID route and the use of 3 different adjuvants with the PEDV vaccine. The second study evaluates the efficacy of ID vaccination using one adjuvant in sows to provide protection to their suckling piglets following wild-type PEDV challenge.
In general, the authors do convey their message to the reader, but there is a lack of details in the manuscript. The lack of details is to an extent where I cannot determine how they made the decisions they did make.
Comment 1. It is not clear what “dose” was used for each adjuvant, and why were they picked the dose, or the adjuvants to be used?
Answer: Three adjuvants are used mainly in Korean pigs, and the three adjuvants were ID inoculated with an inoculation dose of 0.5ml (107.0TCID50/ml). (Revised manuscript line: 219—239 and 286-290).
Comment 2. A decision was made to use the Carbopol 940 adjuvant for the sow study, the adjuvant selection was based on the combination of the Carbopol 940 ID vaccination in pigs generating “the highest NA titer and persistence, and had fewer side effects.” Based on figure 2A, as I understand it, the NA titer of the Carbopol 940 ID vaccination was not the highest or had the longest duration of an elevated NA titer. Unless I missed it, there are no numbers provided to support the author’s claim of less edema following ID vaccination with Carbopol 940 compared to the other two adjuvants.
Answer: Here, we found that the Carbopol 940 polymer (NF) was the best adjuvant for the inactivated PED vaccine because it generated a sufficient and persistent NA titer, with fewer side effects. The anti-PED NA titer and persistence induced by the IMS1313 adjuvant was slightly better than that in pigs injected with the Carbopil 940 polymer adjuvant, but there was prolonged edema at the injection site (lasting for 2 to 3 days). Pigs receiving ID inoculation with inactivated PED vaccine plus GEL01 PR adjuvant showed a lower NA titer and persistence, with prolonged edema at the injection site. (Revised manuscript line:72-78 and 180-186, and revised figure 2A).
Comment 3. I may have missed it as well, but there is no statistical support for any differences in the NA titers or cytokine values for any treatment.
Answer: We added and revised the sentences (Revised manuscript line 72-88 and 324-329, and revised figures legend 2-5).
Comment 4. There is a lack of details on how methods were done, e.g., how were the NA titers determined? No details or reference on how to conduct the test was cited. A reader cannot reproduce the studies in this manuscript because of a lack of details. Is there any real difference between titers?
Answer: We added the NA titer in Material and methods (Revised manuscript line 240-254).
Comment 5. Just looking at Figures 2 and 3, not much evidence that ID is superior to IM. This may be acceptable since part of the goal is to evaluate ID vaccination in sows.
Answer: We removed the IM injection part in the original figures (2 and 3) and revised all sentences and figures (Revised manuscript figure 2).
Comment 6. For the sow study, there is no IM group. The authors do show that the ID vaccination in sows was helpful, it would have been nice to compare the IM to the ID vaccination in sows.
Answer: We added the sentences in Discussion “In future, a commercial inactivated PED vaccine will be compared with respect to its ability to protect suckling piglets via the IM and ID routes. In addition, it is necessary to conduct a comparative experiment that involves oral administration with a commercial live attenuated PED vaccine.” (Revised manuscript line: 210-214).
Comment 7. In final Discussion paragraph the authors speculate ID vaccination will provide good protection if given after an oral vaccination with attenuated PEDV vaccine. I believe this is too much speculation, and not appropriate.
Answer: According to a reviewer’s comment, we removed the sentence.
Comment 8. There is no description of what a mock vaccination was and what a mock challenge was.
Answer: According to a reviewer’s comment, we removed a mock vaccination and a mock challenge in revised manuscript.
Comment 9. Overall, the manuscript read well, but a few phrases or words should be revised. For example, in first sentence of the Abstract, I suggest changing sucking to suckling, and epidemical to epidemic.
Answer: We revised the word (revised manuscript line: 17-18) and this manuscript has been revised by a commercial English revision company (www.bioedit.com).
Comment 10. The Discussion section should be reviewed for precision. In second paragraph, the sentence beginning with “A previous study showed that………. against virus challenge” I am not sure which previous study is being referred to for which pig disease. The citation I could not find without purchasing the book/journal so I do not know what it states. It could be the cited reference has something to do with the general concept of “ID vaccination triggers other immune mechanisms” which may be true, but this comment is very broad and it may reflect some other example in the citation that has nothing to do with pigs, I do not know. I believe other statements in the Discussion may be a bit broad as well.
Answer: According to a reviewer’s comment, we revised the Discussion part entirely (revised manuscript line: 171-214).
Reviewer 2 Report
The manuscript “Efficacy of needle-less intradermal vaccination against porcine epidemic diarrhea virus” was carefully analyzed. The authors described the effect of sow vaccination against porcine epidemic diarrhea virus using intradermal route on suckling piglets experimentally infected. However, no comparison between intradermal and intramuscular vaccination routes is performed in this manuscript as mentioned in lines 57-58.
Introduction:
It should be mentioned that commercial vaccines against PEDV are not available worldwide. It should be remarked the location (i.e. Korea).
If administration routes (intradermal vs intramuscular) are not going to be compared, that part should be removed from the introduction.
Material and Methods:
- The health status of those selected animals (or farm) should be detailed in the manuscript.
- Animal selection (sow and piglets) should be described in this part as well as the distribution per group: How many animals were selected in total? Which parameters were followed? The selection criteria must be included.
- Lines 216-223: Group distribution and number of animals per group are not clearly explained
- A robust experimental design must be done in order to monitor the effect of ID vaccination.
Results:
Preliminary results showed in this manuscript about the effect of sow vaccination against PEDV using ID route are interesting, as could be seen for many pathogens. However, higher “n” must be considered, in particular, if one of the objectives of this manuscript is to compare ID vs IM route.
Discussion/Conclusion:
Results obtained in this study are poorly discussed. Additionally, this study did not evaluate the ID vaccination after oral vaccination with a live attenuated vaccine against PEDV (unknown sow/farm status). Therefore, further studies are needed to confirm the suggestion or expectation proposed by the authors in lines 203-205. Furthermore, the conclusion (L300-302) was not adequate because IM route was not evaluated in sows.
Author Response
Reviewer 2
Comment 1. The manuscript “Efficacy of needle-less intradermal vaccination against porcine epidemic diarrhea virus” was carefully analyzed. The authors described the effect of sow vaccination against porcine epidemic diarrhea virus using intradermal route on suckling piglets experimentally infected. However, no comparison between intradermal and intramuscular vaccination routes is performed in this manuscript as mentioned in lines 57-58.
Answer: We removed the IM injection part in revised all sentences (Revised manuscript line: 56-58).
Introduction:
Comment 2. It should be mentioned that commercial vaccines against PEDV are not available worldwide. It should be remarked the location (i.e. Korea).
Answer: We removed the IM injection part in revised all sentences (Revised manuscript line: 39-44).
Comment 3. If administration routes (intradermal vs intramuscular) are not going to be compared, that part should be removed from the introduction.
Answer: We removed the IM injection part in revised all sentences.
Material and Methods:
Comment 4. The health status of those selected animals (or farm) should be detailed in the manuscript.
Answer: We added the health status of animals. (Revised manuscript line: 224-225, 235-236, 286-287, and 295-297).
Comment 5. Animal selection (sow and piglets) should be described in this part as well as the distribution per group: How many animals were selected in total? Which parameters were followed? The selection criteria must be included.
Answer: We revised the sentence for animals. (Revised manuscript line: 286-293 and 295-300).
Comment 6. Lines 216-223: Group distribution and number of animals per group are not clearly explained
Answer: We revised the sentence of animals. (Revised manuscript line: 217-239).
Comment 7. A robust experimental design must be done in order to monitor the effect of ID vaccination.
Answer: We thanks for the good advices we will refer to it for future research.
Results:
Comment 7. Preliminary results showed in this manuscript about the effect of sow vaccination against PEDV using ID route are interesting, as could be seen for many pathogens. However, higher “n” must be considered, in particular, if one of the objectives of this manuscript is to compare ID vs IM route.
Answer: We removed the IM injection part in revised manuscript.
Discussion/Conclusion:
Comment 8. Results obtained in this study are poorly discussed. Additionally, this study did not evaluate the ID vaccination after oral vaccination with a live attenuated vaccine against PEDV (unknown sow/farm status). Therefore, further studies are needed to confirm the suggestion or expectation proposed by the authors in lines 203-205. Furthermore, the conclusion (L300-302) was not adequate because IM route was not evaluated in sows.
Answer: According to a reviewer’s comment, we revised the Discussion part entirely (revised manuscript line: 171-214).

Round 2
Reviewer 1 Report
The manuscript revisions helps make the overall "story" about your research more understandable for others.
Author Response
Comment 1: The manuscript revisions helps make the overall "story" about your research more understandable for others.
Answer: Thanks for your good advice.
Reviewer 2 Report
The manuscript “Efficacy of needle-less intradermal vaccination against porcine epidemic diarrhea virus” was carefully analyzed. In general, the authors have modified the manuscript, but there is still a lack of details to understand the conclusions according to their proposal.
Lines 25, 155, and so on: Please, change the word autopsy to necropsy (more recommended when you are talking about animals).
Line 63: Please, review the sentence “Pigs were vaccinated with an inactivated PED vaccine via ID route” or change by “Pigs were vaccinated against PED by ID route”.
Lines 70-71: Figure 1B and 1C are not mentioned/cited in the text.
Lines 170-214: The discussion and conclusion section must be improved because only results are described (it seems a result section). Obtained results in this manuscript should be compared/discussed with previously published other viruses as SIV or HIV. Few references (26-29) were used to discuss these interesting results. Discussion is the most important part of an article.
Lines 215-330: Materials and methods are not clearly explained in this manuscript. This section should be improved in order to understand the trial design: adjuvant selection + efficacy of ID vaccination. Furthermore, there is not clear the animal selection based on. Which is the farm/selected animals health status regarding the main swine diseases? (i.e PRRSv, PCV2, …) or pig selection was based on just apparently healthy? Please, add it because depending on the basal status, results could vary a lot.
Author Response
Comment 1: The manuscript “Efficacy of needle-less intradermal vaccination against porcine epidemic diarrhea virus” was carefully analyzed. In general, the authors have modified the manuscript, but there is still a lack of details to understand the conclusions according to their proposal.
Answer: According to reviewer’s comment, we revised and added in discussion and in material & method (revised manuscript lines 180-185, 197-204, 215-220, 239-247, and 314-315).
Comment 2: Lines 25, 155, and so on: Please, change the word autopsy to necropsy (more recommended when you are talking about animals).
Answer: We changed autopsy to necropsy (revised manuscript lines 25 and 155).
Comment 3: Line 63: Please, review the sentence “Pigs were vaccinated with an inactivated PED vaccine via ID route” or change by “Pigs were vaccinated against PED by ID route”.
Answer: We changed “Pigs were vaccinated with an inactivated PED vaccine via ID route” to “Pigs (50-days-old) were vaccinated against PED by ID route” (revised manuscript lines 63).
Comment 4: Lines 70-71: Figure 1B and 1C are not mentioned/cited in the text.
Answer: We added the Figure 1B and 1C in the text (revised manuscript line 74 and 254-256).
Comment 5: Lines 170-214: The discussion and conclusion section must be improved because only results are described (it seems a result section). Obtained results in this manuscript should be compared/discussed with previously published other viruses as SIV or HIV. Few references (26-29) were used to discuss these interesting results. Discussion is the most important part of an article.
Answer: According to reviewer’s comment, we revised the sentences in discussion (revised manuscript lines 180-185, 197-204, and 215-220).
Comment 6: Lines 215-330: Materials and methods are not clearly explained in this manuscript. This section should be improved in order to understand the trial design: adjuvant selection + efficacy of ID vaccination. Furthermore, there is not clear the animal selection based on. Which is the farm/selected animals health status regarding the main swine diseases? (i.e PRRSv, PCV2, …) or pig selection was based on just apparently healthy? Please, add it because depending on the basal status, results could vary a lot.
Answer: According to reviewer’s comment, we revised the sentences in material & method (revised manuscript lines 239-247 and 314-315).